# The Influence of Research and Innovation Strategies for Smart Specialization (RIS3) on University-Industry Collaboration

Carla Mascarenhas [1,*], Carla S. Marques [1], João J. Ferreira [2] and Anderson Rei Galvão [1]

[1] CETRAD Research Center, University of Trás-os-Montes e Alto Douro, 5001-801 Vila Real, Portugal; smarques@utad.pt (C.S.M.); anderson@utad.pt (A.R.G.)
[2] NECE-UBI Research Unit in Business Sciences, University of Beira Interior, 6201-001 Covilhã, Portugal; jjmf@ubi.pt
[*] Correspondence: carlam@utad.pt

**Abstract:** This study aims to understand better research and innovation strategies for smart specialization (RIS3) and assess how they influence university-industry (U-I) collaboration empirically. Primary data were collected from a focus group consisting of representatives of universities and government entities from Portugal and Spain. Secondary European Union (EU) data on the application of smart specialization measures and the innovation in these two countries were also included. The results reveal difficulties in implementing RIS3, resulting in decreased investment in research and innovation in all production sectors. Evidence also pointed to the negative impact of smart specialization measures on U-I collaboration and, consequently, on the respective knowledge transfer.

**Keywords:** smart specialization strategy; RIS3; entrepreneurial discovery process; governance

## 1. Introduction

The global crisis that began in 2008 brought European Commission (EC) attention to the situation in some of its member countries and exposed key problems and unsustainable developments in many member states [1]. The sluggishness in productivity recovery and the financial crisis have underlined that EE countries need a renewed emphasis on obstacles to structural change and well-functioning innovation systems [2]. The effectiveness of research systems plays an important role in economic growth and directly impacts all sectors of current society. The level and efficient use of public research and development (R&D) funding are decisive factors, along with the establishment, maintenance and development of mechanisms to improve structural performance, conditions and processes [3]. Therefore, as part of its cohesion policy, the EC devised the Smart Specialization Program to identify strategic areas of intervention based on analysing the economic strengths and opportunities in each region. This innovative approach aimed to boost growth and jobs in Europe and enable each region to identify and develop its competitive advantages [4].

The academic concept of smart specialization was developed in the mid-late 2000s [5]. Afterward, the EC's Knowledge for Growth Expert Group was chosen to develop this idea and make it a reality [6,7]. Specialization in R&D and innovation is crucial, especially for the less technologically developed countries [6]. According to the European Growth Strategy, the EC aims for the European Union (EU) to "become a smart, sustainable and inclusive economy" by 2020. These three mutually reinforcing priorities should help the EU and member state employment, productivity and social cohesion [8]. To this end, five targets have been set for 2020 (e.g., employment, innovation, education, social inclusion and climate/energy), and each member state has adopted its national targets for each area. Thus, national and regional authorities across Europe have devised smart specialization strategies so that the European structural and investment funds can be used more efficiently and synergies between different EU, national and regional policies might be increased in conjunction with the levels of public and private investment [8].

Thus, more than 120 smart specialization strategies have been developed by member states and their regions through partnerships, multilevel governance and bottom-up approaches, setting priorities for research and innovation investments for the 2014–2020 period. That is, public funds to support research, innovation and development will be applied to the best projects that are strategically aligned with the priorities set in each country and/or region [9].

The detailed analysis of each region, its innovation index, and its priority investment objectives has brought about a better and more thorough understanding of the regional economies. This, in turn, led to the design of support policies tailored to each region. However, the effect of these smart specialization regional development policies and the Entrepreneurial Discovery Process (EDP) need to be assessed [10,11]. Moreover, understanding the impact of these measures on the relationship between universities and industry is also essential. Given this gap in the literature, this study aims to elucidate the smart specialization strategy and empirically evaluate these measures' influence on U-I collaboration. Thus, this research is based on the following research questions:

- What is the impact of RIS3 on the U-I relationship?
- How has RIS3 influenced U-I knowledge transfer?

The article is structured as follows. The next section presents the literature review, and the third one describes the methodology. The main results appear in the fourth part. A discussion of the results follows before the last section presents the conclusions and policy implications and suggests future research lines.

## 2. Literature Review

### 2.1. Regional Innovation Systems

One of today's greatest challenges is generating, applying and disseminating scientific knowledge and transforming it into technological innovation. The growth of countries such as the United States, Germany and Japan demonstrates how a favourable national environment can influence stimulating innovation. When the US and European economies were losing out to Japan, the latter became a major world leader [12]. Freeman [13] and Lundvall and Andersen [14] have highlighted the differences in the rate at which countries exploit the possibilities offered by technological gaps, especially in those periods when major changes occur in technical-economic paradigms or technological trajectories. These differences are believed to depend on each country's ability to mobilize political and financial resources to transform the technological, institutional, and economic structures that comprise the national innovation system [15]. The debate on industrial policies in the US and Europe in the 1980s led to the concept of the National Innovation System [16] that has since been widely studied and defended by [17–19].

The economic crisis of 2008 led to the creation of the EU smart regional specialization strategy (RIS3) that has aimed to make European regions internationally competitive [20,21]. Given each member state's specificity and heterogeneity, smart specialization seeks to bridge the gap between less innovative regions and technology centres by identifying each region's unique innovation resources. Each has had to carefully analyse its strengths and weaknesses and determine the areas to invest in becoming innovative and competitive [22]. This allows the regions to include those sectoral projects with the greatest promise to promote regional development and continuously seek out new and innovative activities [23–25]. This context led us to the first proposition (P1) of this study:

**P1:**   *Smart specialization encourages innovation in countries considered less innovative.*

The smart specialization strategy's main purpose is to modernize traditional segments, being progressive by definition [22,23,26,27]. Thus, the second research proposition (P2) was formulated:

**P2:**   *Choosing priority investment areas for innovation facilitates the transfer of technology and knowledge through funded projects.*

However, smart specialization measures require high governance capacity levels, especially in regions with limited institutional resources, such as economically weaker regions. Existing studies show the need for an explicit and structured process of analysis, reflection, and prioritization within the chosen domains and areas [28], in addition to the monitoring and evaluation of activities, which must be adapted to each regional context [24,29]. Thus, the third proposition (P3) was:

**P3:** *The governance models process is successfully implemented, facilitating the necessary changes to RIS3.*

With the present community framework in its final stages, it is necessary to study and clarify the measures adopted to be reviewed within the next investment cycle. Several studies point to the need to review the priorities and themes chosen [29,30]. Taken together, the socio-economic analyses of the European regions studied herein may help to clarify whether priority changes for each regional and national smart specialization strategy are indicated.

### 2.2. Entrepreneurial Discovery Process

Dominique Foray states that RIS3 is "largely about the policy process for selecting and prioritizing fields or areas if a cluster of activities is to be developed and allowing entrepreneurs to discover the right domain of future specialization" [31]. To identify the areas to receive potentially transformative investments, the RIS3 concept suggests that an entrepreneur-driven, self-discovery process will be implemented better to understand regional opportunities [32,33].

The smart specialization approach refers to how competitive advantage can be uncovered using EDP. However, RIS3 does not refer to entrepreneurship as commonly described, rather creating a company as an individual entrepreneurial project [34,35]. Instead, in this context, entrepreneurship should be more broadly understood to encompass all stakeholders supporting entrepreneurship, whether they be entrepreneurs, other companies, universities or other higher education institutions, as well as government entities and society in general [36]. These actors can discover innovative domains to guarantee existing and future competitiveness and entrepreneurial discovery [37]. The role of different stakeholders varies in terms of involvement and impact on said discovery. Therefore, the process needs to be business-oriented [38]. From this arose the fourth research proposition (P4):

**P4:** *The EDP uncovers new opportunity domains.*

Entrepreneurial discovery serves as the main information source for exploring new opportunities and the transformational activities that must be prioritized. The private sector should be managed with the support of research and innovation institutions, while governments will ensure successful implementation and enhance actor coordination [23,39].

Thus, regional policymakers aim to foster a region-based EDP that might generate intensive, customer-centred discovery. This direct addition of end-users to the innovation process is a necessary organizational counterpart to an open innovation policy in that it accords greater attention to consumers' underlying needs. This quadruple helix approach allows for a wider range of innovation, in addition to that based on technology or science [1,40].

Therefore, smart specialization rejects the "pick winners" culture, favouring a radically new process of financial support for innovation. The new strategy requires public-private partnership in entrepreneurial discovery, and it is, thus, implemented as a bottom-up process of the self-discovery of entrepreneurial capacity [41]. Recent studies have recognized that the process must incorporate both ascending and descending processes [42,43], thereby establishing broader and regionally more favourable research priorities [44].

Figure 1 illustrates the concepts discussed and this study's research propositions.

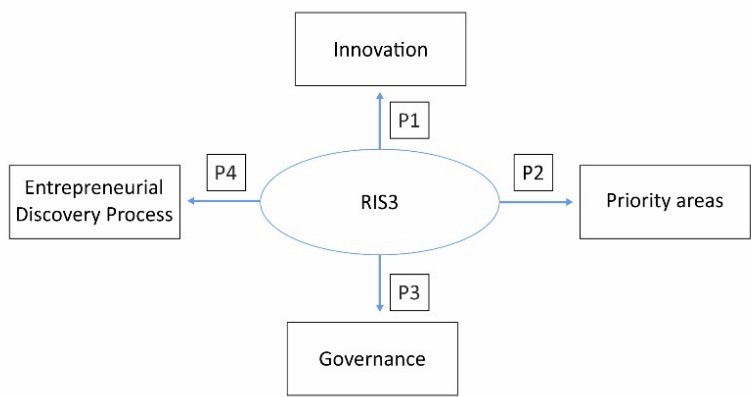

**Figure 1.** Research Model.

## 3. Methodology

Since the objective is to identify the critical perception of how RIS3 influences U-I collaboration, its benefits, and any impediments, a qualitative study using interviews and focus groups was the most appropriate research method.

Focus groups, originating from sociology [45] and initially used in marketing research [46], have gained currency in social science research. This methodology can be used to elucidate a spectrum of perceptions and attitudes about a fact, practice, product, or service. It is similar to a group interview, without formalizing questions and answers between researchers and interviewees [47,48]. Some researchers argue that focus groups' interpersonal and interactive nature collects information that might not be obtained from a single interviewee, thus generating a wider range of insights and ideas [49–52]. This focus group comprised eleven representatives, five universities (three Portuguese and two Spanish) and six regional governing bodies (four Portuguese and two Spanish).

Information was collected from each region's RIS3 documents, statistics from the Regional Innovation Scoreboard, PORDATA, EC reports and information available on official pages to triangulate the data.

This data included R&D expenditure in the public and business sectors, Non-R&D innovation expenditures [53–56], as well as Expenditure in R&D and researchers involved in R&D activities (https://www.pordata.pt/en/DB/Europe/Search+Environment/Chart; https://www.pordata.pt/en/DB/Europe/Search+Environment/Chart, accessed on 10 Feburay 2021).

The data (documents, interviews, and focus group) was subjected to content analysis. By organizing sources and coding data with NVIVO 11.0 software, it was possible to extract useful and segmented information and create a tree map that labelled and branched the data. When the information was too varied in content or lacked sufficient instances to capture patterns, thereby leading to coding difficulties, word cloud analysis was used. This paper critically analysed the RIS3 implementation and how said measures might have influenced U-I collaboration. The focus group opinions were transcribed to facilitate their reproduction and analysis.

## 4. Results

The focus data was sorted and categorized using various codes and categories. These codes can either be either loose or grouped (into categories). Similar codes grouped related data, and novel information prompted new categories. The most commonly discussed topics were the RIS3 implementation itself, governance models and the EDP. Figure 2 shows participant opinions about influences on the RIS3 process.

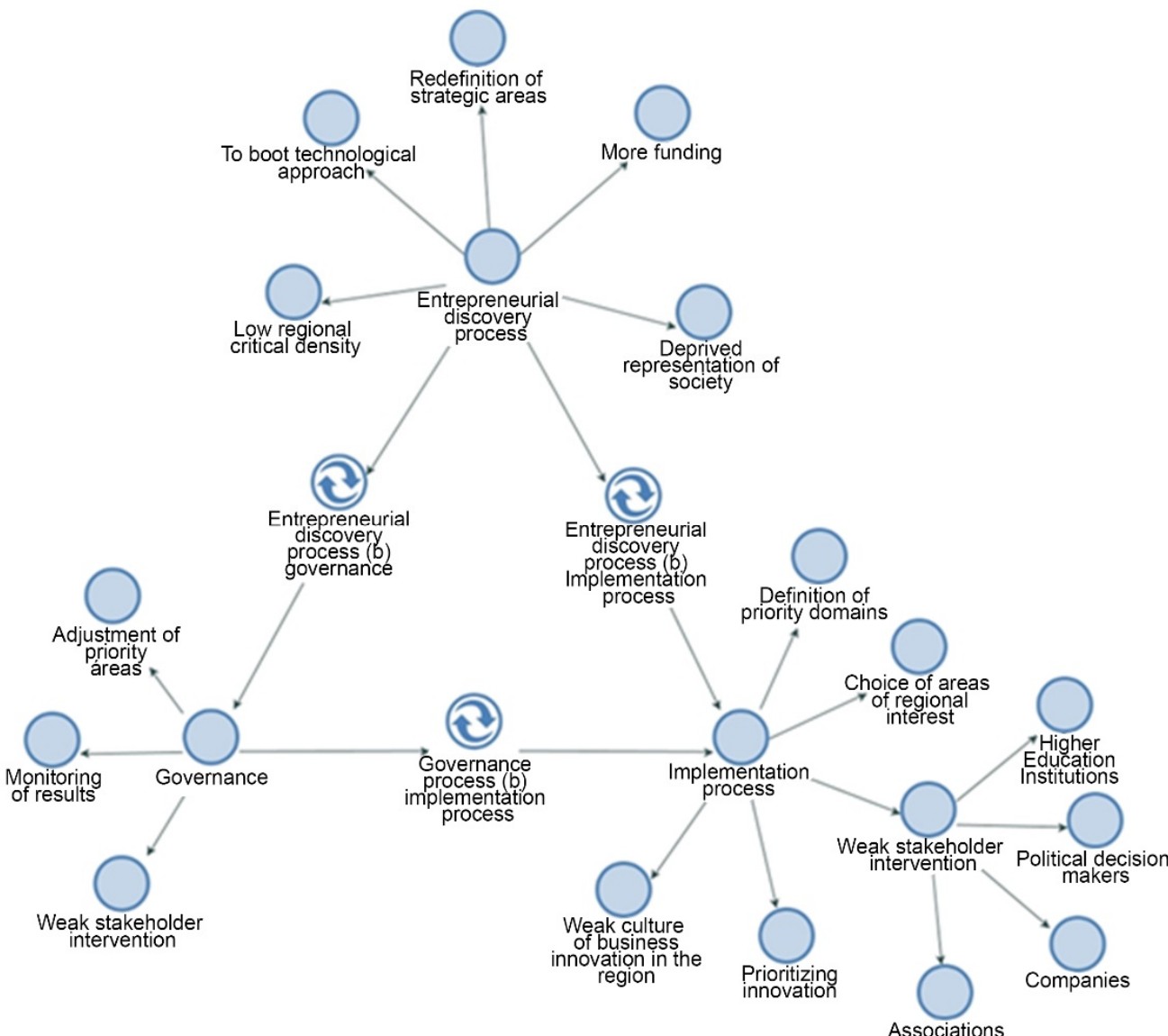

**Figure 2.** NVivo Node Map and Relationship.

RIS3 is shown to be associated with its implementation process and, therefore, with the entrepreneurial discovery and governance processes. Participants recognized that these factors might put the smart regional specialization process at risk in Portugal and Spain. In addition, the RIS3 implementation was shown to be closely associated with the choices of priorities, feeble stakeholder intervention, prioritization of innovation and insufficient entrepreneurial culture in the region. Weak stakeholder participation in the governance process was also seen as an obstacle. Other influences included changes regarding priority areas and results monitoring. EDP, in turn, was influenced by a redefinition of strategic areas, RIS3's overly-technological approach, weak regional critical density, as well as insufficient societal representation and the need for greater business support.

The following section compares Portuguese and Spanish innovation and uses respondent quotes to illustrate the three main nodes of Figure 2.

### 4.1. P1: Smart Specialization Has Spurred Innovation in Countries Considered Less Innovative

Portugal and Spain were, until 2019, very similar in terms of innovation, both being "moderate innovators", with innovation performance decreasing between 2010 and 2016 compared to the EU average [53]. These data reflect that these countries' entrepreneurial

fabric consisted mainly of micro and Small and Medium Enterprises (SMEs), with low absorption and innovation capacity. The percentage of highly innovative companies is extremely low. For example, Portuguese levels of innovative, intramural SMEs decreased in 2016 to 79% of the EU average [54]. The R&D/GDP business expenditure ratio has also declined recently, reaching a low of 1.27% in 2016, after a previous historical low of 1.58% in 2009. This reveals a lack of concern with R&D investment in the business sector and all execution sectors.

In 2020, Portugal became, a "strong innovator" [57]. The strong increase registered in 2018 is explained by the exceptional performance obtained in indicators such as "Innovators", "Innovation-friendly environment," and "Attractive research systems". According to the European Innovation Scoreboard 2020, "Portugal scores particularly well on SMEs innovating in-house, Broadband penetration, SMEs with product or process innovations, and Foreign doctorate students. Sales impacts, Linkages, and Intellectual assets are the weakest innovation dimensions. Portugal's lowest indicator scores comprise Exports of knowledge-intensive services, R&D expenditures in the business sector, Private co-funding of public R&D expenditures, and Public-private co-publications" [57].

In Spain, lower relative spending on business research and development (€ 7 million versus an average of € 29.9 per 10 million inhabitants over the period 2011–2015) is an obstacle inhibiting innovation. Over the last two years, the rising Spanish GDP growth rate (3.4% in 2015, 3.3% in 2016) has not translated into greater innovation. R&D intensity has continued to decline since 2010 [55], suggesting an unwillingness to invest in R&D across all sectors.

As regards Spain, the European Innovation Scoreboard 2020 strongest innovation dimensions are "Human resources", "Innovation-friendly environment" and "Employment impacts". "Spain scores high on New doctorate graduates, Sales of new-to-market and new-to-firm product innovations, Broadband penetration, and Population with tertiary education. Innovators, Firm investments, and Linkages are the weakest innovation dimensions. Low-scoring indicators include Exports of knowledge-intensive services, SMEs innovating in-house, SMEs with product or process innovations, and R&D expenditures in the business sector." [57]

When looking at the percentage of jobs in knowledge-intensive service sectors, Portugal's gap relative to that of the EU is even greater (29.6% versus the 2016 EU average 37.2%), while the high and/or medium production sectors and high technology also lagged behind (2.33% and 4.63% in 2015). Spain, as well, has fewer jobs in knowledge-intensive services (51.2% in Spain versus 58% in the EU from 2011 to 2015) and in high and/or medium technology production (31% of employment) than the European average [56].

These figures reflect changes in both financing patterns and R&D performance, notably RIS3.

There is little Portuguese data available about RIS3 implementation. However, all regions have published their priorities and appear to be well prepared to fully implement the process, while major limitations remain regarding levels of entrepreneurial discovery implementation [56].

All 17 Spanish regions adapted their research and innovation to smart specialization (RIS3) in 2017, many of which identifying similar priorities. However, intergovernmental coordination is less than ideal, resulting in a lack of optimal synergy between national and regional policies [58]. In Spain, as in Portugal, many entrepreneurial discovery implementation issues remain [55].

*4.2. P2: Choosing Priority Investment Areas for Innovation Has Facilitated the Technology and Knowledge Transfer through Funded Projects*

In 2010, partly because of the devastating economic crisis in Europe, the continent faced and continued to face major economic challenges that require an ambitious 21st century economic policy. The EU 2020 strategy was to foster a social market economy, intended to address structural weaknesses by making progress towards three mutually reinforcing objectives: (i) smart growth, based on knowledge and innovation; (ii) sustain-

able growth, promoting a more resource-efficient, greener, and more competitive economy; and, (iii) inclusive growth, enhancing a high employment economy, while also providing economic, social and territorial cohesion [59]. Thus, each country and region has identified priority areas for applying community funds, considering their economic and technological environment and identifying their competitive advantages. However, contrary to EC guidelines, the choices were not standardized, as respondents pointed out.

> *[In Portugal, some regions have narrowed down their priorities to such an extent that important projects are excluded. These are great value projects, some even disruptive, but funded they have not been funded due to a lack of a certain wording. For the most part, in Spain, most of the priority areas are too transversal, and they permit almost anything. No serious, coherent and critical investigation has taken place into the regions. Again, this is a European problem, not just limited to Portugal or Spain.]* (Portuguese government participant, PG).

> *[There are regions, in Spain and Portugal, with too many domains. I don't understand how projects with so many domains can be funded. And many of them are transversal. There is no intelligent specialization in the true sense if we look closely, neither specialization nor intelligent. Look carefully at smart specialization strategies in Spain or Portugal. There are similar domains in all of them. In fact, smart domains were not chosen where the regions had proven critical capacity. Areas were chosen, period. This is the problem of the least innovative, least innovation-critical countries, with a business fabric composed mostly of micro and SMEs.]* (Spanish government representative, SG).

Capello and Kroll pointed out as early as 2016 the lack of regional and/or national interest, low innovation capacity, and general policies for RIS3 implementation across many European regions [41]. The methods for determining and implementing priorities were called attention to by the focus group:

> *[Little progress has been made in the granular process of choices. The EU made the choices, and the countries or regions accepted them, and it is, therefore, easier to say that they were made by someone other than us. We looked macroscopically at the areas of regional interest. This leads to projects being funded only because they fall under RIS3, leaving important areas outside]* (PG).

One of the difficulties underlying the smart innovation specialization process was the way the stakeholders were heard. Indeed, any innovation system needs different actors participating in a complex network of relationships and dependencies, and innovation policy must appropriately address this complexity. All players in the innovation system should be heard. However, due to the urgency of combating an ongoing economic crisis, European smart specialization policies were implemented in a hurry, if not totally "recklessly" [60]. All participants were not appropriately consulted. By way of example:

> *[Some European regions already had innovation policies similar to smart specialization, so it was nothing new. In these regions, the RIS3 works like a well-oiled machine. For others, measures have become mandatory, giving rise to problems. In the latter, stakeholder involvement was, and still is, almost non-existent. In Portugal, for example, universities played a very weak role, just listening and accepting the proposals by regional leaders. The few choices made were the easiest. Little input was sought out from academics. The same was true of businesspeople. There was no articulation between universities and companies. Policymakers did the work and the same has taken place in many other European regions]* (PG).

These ex ante problems have led to several smart specialization implementation challenges. Lesser-developed countries and/or regions often lack the formal institutions to support these new measures and face serious problems in implementing them [28].

> *[RIS3 should have been a continuous learning process. But much remains to be done Smart specialization cannot be a mere prioritization process. Innovation cannot be blindly prioritized. There are a number of extremely innovative projects that are not funded*

*solely because RIS3 does not cover the areas where they are located. There are innovative products that will probably never reach the market because companies have neither the financial capacity nor the knowledge to carry them forward]* (Portuguese university representative, PU).

*[How to guarantee the mechanisms of choice? RIS3 should allow regions to choose which projects they consider fundamental for their development, but we now know that they are chosen because they meet the conditions of admissibility of RIS3 and merit rather than the strategic dimension that was intended. Moreover, for that to occur, we would need deep technical work]* (Spanish university representative, SU).

Indeed, to take advantage of future EU opportunities, three types of strategic skills are needed: the ability to identify local forces align political actions, and create critical mass; and the regions' capacity to develop a vision and implement the smart innovation strategy [61].

### 4.3. P3: Process Governance Models Are Properly Implemented, Facilitating Necessary Changes to RIS3

RIS3 requires a combination of integrated policies beyond R&D and that require broader "transformation policies" at different levels. Therefore, governance models are needed to ensure a coherent policy mix to support smart specialization strategy (S3) priorities. However, efficient communication is often lacking between national governments. This implies that, instead of being a purpose-driven construction, existing policy combinations are often the unintended product of an accumulation of instruments over time and across policy domains and levels [8].

The RIS3 process should be interactive and region oriented. While the precise organizational mix depends on the regional context, all partners must be fully involved in developing, implementing and monitoring smart specialization strategies [59,62].

*[In addition to ex ante problems such as those mentioned above, the implementation of RIS3 also faces difficulties in governance models. Respondents complained about an almost total absence of such models, despite being mandatory. In the words of one participant, "In most regions, and in most cases, the involvement of all partners, institutions and actors is extremely limited. Here governance models begin to fail"]* (SU).

Another participant mentioned,

*[We can't mobilize partners for innovation, discovery, and problem-sharing meetings when these people think we're stealing time, because they don't understand what they'll get when they go there. This has to be demystified. Stakeholders have to understand what their role is in this smart specialization process. It has to be dynamic. What was a wild card when originally choosing areas may no longer be so? Seven years have gone by, and innovation does not appreciate such long periods]* (PG).

*[Unlike my Portuguese colleague, I think that we can mobilize the actors, the stakeholders, whoever they are, if the meetings, the focus groups, the workshops are thematic, if concrete issues are discussed, concrete problems, not RIS3 or its themes as a whole]* (SG).

Awareness of strategy policy is, therefore, an essential tool for smart specialization governance. However, in practice, the link between policy instruments and priority setting is not explicit in most regions and countries [61]. To this end, benchmarks and criteria for success and failure are needed. Policymaking should be flexible enough to discontinue or reallocate public resources when assessments show that the goals are not being met [10,24,29,59,63].

The absence of true governance models has, thus, brought about several obstacles to implementing smart specialization. One Portuguese university representative expressed the opinion, "As there is no governance model, there were Portugal adjustments at either the regional or national RIS3. There was no adjustment of the strategies, which have remained in the drawer." A Portuguese government official, meanwhile, said,

*[Governance models are not implemented because, in RIS3, what matters is that funding is allocated according to the rules imposed. Only when Brussels enforces it that governance models will appear and will be accorded their due importance, when the Community Board is about to finish and if they have to present accounts to play on the Community Framework 21–27]* (PG).

Additional governance challenges or the absences of RIS3 governance mechanisms, such as creating two-way communication channels include a shortage of qualified staff in agencies and ministries. The latter is especially marked in the lesser developed and more remote regions, especially when facing constraints on public finances and public sector employment [24].

This was one of the issues raised by the focus group that partly justifies the absence of governance models, in the words of a Spanish government representative, *["The problems with a lack of governance or monitoring models have to do with the public sector capacity limitations, and the paucity of financial resources, personnel and qualified personnel for the process."]*

Governance models need to be monitored to assess whether measures are being implemented at the regional level. Various support tools are available. This is critical, especially regarding the adoption of composite indicators during the weighing and aggregation phases [30,64]. In the words of one participant,

*[There are no governance models because there is no real monitoring of results, no monitoring of indicators of realization, timing, allocated funds, even less in terms of real economic benefits for the region, or innovative market results. What is the economic result for the company? The EU appeals to the value of money, but the regions lack effective indicators of said value. What is the return on funds allocated? Moreover, this problem is not new; it is perpetual]* (PG).

A Spanish government participant stated, *["We have several RIS3 instruments for monitoring and governance. What is lacking is their implementation. So, we are very late, not only in Spain or Portugal, but throughout Europe" (SU.)]*.

Smart specialization strategies can, however, clash with other regional innovation policy objectives, undermining EDP [41,65]. Therefore, regional governance systems need structural change support strategies that focus funding on measures capable of yielding the greatest return on economic and social development [66]. A Spanish university representative stated that the lack of effective and efficient governance models may undermine the RIS3 process, *["The designed model (RIS3) is interesting. However, governance models must have room to intervene and function. We have to find room for entrepreneurial discovery. This is not possible without a good governance model."]*

### 4.4. P4: New Opportunity Domains

Knowledge spillovers are undoubtedly one of the essential elements of intelligent specialization theory [27]. Thus, RIS3 is the regional economic structure that encourages investments in R&D and innovation, and that should enable it to generate scientific, technological and economic expertise, thereby increasing competitiveness, productivity, and, ultimately, economic growth [67].

The smart specialization approach requires a so-called "entrepreneurial selection" of market opportunities that minimizes failures and avoids policy errors. In practice, this means promoting entrepreneurship in general. While successful companies constitute the new country and/or region specialization (self-discovery), the policy framework goal is to develop a flexible strategy that focuses on measurable intermediate objectives, identifying constraints and market failures, and ensuring feedback on learning processes [61]. What distinguishes smart specialization from traditional industrial and innovation policies is undoubtedly the EDP. However, in respondents' view, the process has been delayed and is almost non-existent in many regions.

*[The entrepreneurial discovery process is slow, as might be expected. It started with a very technological approach. There have been various criticisms of this approach, and*

*Europe has already become aware of this problem, and now it is saying that the process should not be so technological. And that's why entrepreneurial discovery is still far from what it should be and from what was set out in the EC's guidelines]* (PG).

*[There is no entrepreneurial discovery in the true sense that RIS3 wants it to exist. Incidentally, it does not exist at all. Choices were made in areas considered strategic, and companies are now intended to innovate in these areas, in so-called golden niches]* (PU).

Indeed, the main obstacles to knowledge-based growth in lesser industrialized or developed regions include a dearth of highly skilled human capital, the lack of developed entrepreneurial culture, and low absorptive capacity levels [68]. The limited sustained fit between the fourfold propeller of innovation stakeholders is also an issue, particularly between industry and academics [1]. Most SMEs, particularly in less developed regions, are microenterprises with few resources to devote to research and innovation and highly dependent on their regional innovation ecosystem. RIS3 aims to engage SMEs in this ecosystem, increasing their innovation potential, providing better access to financial resources and human capital [69].

*[There are several restrictions on science. Most companies in the lesser developed regions, whether Spanish, Portuguese, or any other European country, are microenterprises, or SMEs, which lack the critical mass needed to succeed internationally. They cannot innovate on their own. Here, universities and policymakers have failed. Companies need support to find the difference, the "entrepreneurial breakthrough". They cannot do it alone, however much they hope or try to]* (SG).

*[I think that, in most of the regions, there are some misconceptions. What is the EDP about? The same name is used for processes that must be very distinct. When we have meetings around the themes chosen at RIS3, we may call them entrepreneurial discovery, but they are not effective. The entrepreneurial breakthrough should sit the quadruple helix players at the same table in search of new projects, starting from innovations of interest to a particular cluster or company]* (PG).

While smart specialization implies a growing focus on economically promising sectors, the formulation and implementation of RIS3, centred on the effective EDP, may require the identification and subsequent exploration of the potential for significant change. A shift to more competitive and specialized production may require regional stakeholders to look for radical products or process developments or even both [29]. This is an interactive process in which all stakeholders come together for a more innovative, economically more developed region [32,70].

However, some respondents believe that universities should lead the EDP by promoting the reformulation of existing priority domains as centres of knowledge and innovation.

*[Most regions lack the critical density to meet challenges. Here, universities can and should play a key role. As centres of knowledge, science, and innovation, they must move out of their comfort zone and look for market challenges. This is even more complex in regions where there are no challenges. Most higher education institutions should not be sitting and waiting for the next era, waiting to see their role in that new period. They should already be involved in the smart strategy readjustment intervention process]* (PG).

Many policymakers find it difficult to move from prioritization to policy instrument development and the corresponding budgets. In most cases, prioritization is disconnected from the budgetary process [71]. This is one of the obstacles to the EDP, and, as such, of the entire smart specialization strategy [37].

## 5. Discussion: RIS3 and Open Innovation

### 5.1. Discussion: RIS3, and University-Industry Collaboration

Since 2014, access to European structural and investment funds for innovation has been determined by the priority areas chosen by each of the regions according to intelligent specialization. With regional RIS3, local actors' influence in policy formulation directs

the approach [72,73]. This can make it difficult to implement and govern these measures, particularly in the lesser developed or less innovative regions [24,30].

Results show that Spain's innovation rate has not changed, contradicting proposition 1 (intelligent specialization stimulated innovation in countries considered less innovative). Portugal, in turn, has gone from moderate innovator to strong innovator. In this case, the proposition has been validated.

As the community framework has almost reached an end, and after applying European standards for implementing a smart specialization for innovation, Portugal and Spain's results are not positive. If we look at the results of the Regional Innovation ScoreBoard 2020 in terms of innovative products and processes, although some regions in Spain have shown an increase in this indicator, the majority have obtained worse or similar results to 2017 [74]. The non-validation of proposition 1 seems to be in line with the results obtained by the European Commission and the Spanish Government regarding the evaluation of the implementation of RIS3 in this country. According to [75] there have been several constraints to implementing smart specialization policies in Spain.

Some of these *ex-ante* constraints include:

- The overlap between the closure of the 2007–2013 period and the first years of the 2014–2020 period has led to a delay in implementing the Operational Programmes for this period and, therefore in the start-up of RIS3. Likewise, there was insufficient capacity to budget for actions from two different periods in many regions, and credits from 2014 and 2015 have been used to close the previous 2007–2013 period.
- The delay in implementing the 2014–2020 period due to the delay in the European publication. In addition, until well into 2015, a large number of delegated acts that impacted the development of the operational programs continued to be produced; for its part, the national regulations were also delayed, and the eligibility order was not published until 2016, and there was also a delay in the approval of the funds and the adoption of the operational programs.
- The increasing complexity in the management and justification of ERDF funds in this period. This difficulty has created, among other things, the designation of Intermediate Bodies, the difficulties encountered by beneficiaries when implementing actions, the lack of definition or changes in eligibility criteria, etc.
- The participatory governance for elaborating the RIS3 has led to the proposal of new instruments and actions in line with the new thematic areas and objectives pursued. However, implementing these new instruments is limited by the existing regulatory framework, the administrative procedure, and the existing inertia in the current systems.
- The slowness in the administrative processing of procedures.

Regarding the constraints felt during the RIS3 implementation, the evaluations reports point to:

- The RIS3 exercises in Spain showed a slipshod prioritization, i.e., many specialized areas than each regional economic structure can justify.
- Although entrepreneurial discovery is at the heart of some strategies (they have been included as part of participatory governance processes) the actual (or at least operational) integration is not clear.
- In general, policy instruments and measures still lack specific (and tailored) approaches to meet the specific needs of sectors and innovation. sector-specific and innovation-specific needs: they are horizontal and rather traditional concerning past periods.
- Even though evaluation design and monitoring efforts have increased considerably, it is unclear how complexity (and even general measures) can contribute to better policy setting over the whole implementation period.
- RIS3 definition exercises have probably led to very formal documents, and risked preventing the objectives proposed being met (due to lack of resources, capacities, expectations, etc.).

As regards Portugal, the number of innovative products and processes has been increasing since 2017 [74], a period in which the results of the implementation of RIS3 measures have been felt concerning the Community funds allocated at the start of the European Community Framework (Horizon 2020).

The RIO Country Report 2017 pointed out that Portugal, in particular, had one of the lowest investment rates in the EU. Increased funding for R&D remained insufficient to update the national research and innovation system. Low-worker qualification level hinders investment and productivity growth. Portugal remained specialized in low and medium-low technology sectors, with multiple challenges restricting the ability to explore knowledge-intensive sectors. Indeed, although in 2020 it became Strong Innovator, the employment in medium and high-tech manufacturing and knowledge-intensive services only grew in two of the seven regions [74]. Indeed, a study of [76] shows that only large companies in Portugal consider work carried out in collaboration with universities as important. As the countries under study (Portugal and Spain) consist mostly of micro and SMEs, the implementation of RIS3 measures can be questioned.

In Spain, several factors affecting productivity, such as underinvestment in public and private R&D, and poor coordination at all levels of governance, and insufficient policy evaluation hampers innovation. Stronger cooperation between academia and business is needed. Only in this way can we contribute to knowledge diffusion, increasing the number of innovative products and processes [73,75]. Indeed, both countries' university entrepreneurial capacities still need to be improved, translating into limited regional innovative development. Thus, considering restrictions on European funding imposed by the priority domains selected for each region, RIS3 can be seen to have had a negative impact on the U-I relationship. Business innovation is heavily dependent on funding, and companies cannot innovate Intramural. If they are not aligned with set priorities, they will not be able to compete for community funds and will not seek collaboration with universities. If there is no U-I collaboration, knowledge transfer is likewise compromised. Thus, research proposal 2 (P2: The choice of priority investment areas for innovation facilitated the transfer of technology and knowledge through funded projects) seems to be contradicted by this study. In Portugal, the analysis carried out by Nacional and Regional priority areas shows that the fairly broad set of Nacional priority areas corresponds in terms of approved eligible investment to a strong concentration on three or four areas for the set of typologies of operations analysed. At the national level, this degree of concentration is lower, with a relatively limited number of priority areas appearing with less notoriety in terms of approved eligible investment. In general, the types of operations in which the national and regional guidelines are a simple criterion of merit allow some National priority areas to show a presence that they cannot obtain in the types in which these guidelines are an admissibility condition.

Smart specialization is based on the premise that identifying the right priorities for a given territory must involve all regional stakeholders (business, government, university, and civil society). The interaction between all these actors, the fourfold helix, must be enterprising and targeted at transforming the economy via innovation. This requires sophisticated governance capable of involving all these actors, which goes beyond traditional governance forms typically led by one actor (usually a governmental entity) [61,77]. Advances in productivity and competitiveness require prioritization, thus requiring inclusive governance that enables the exploitation of knowledge and the alignment of agent all local agents' capabilities. According to the focus group, most Portuguese or Spanish regions have no governance model in place, and there have been no changes to RIS3. These arguments contradict proposition 3 (Process governance models are properly implemented, facilitating necessary changes to RIS3).

One of the conclusions of the evaluation report in Portugal is the recognition that in Regional RIS3 the virtuous relationship between the characteristics of regional innovation systems (RIS) and the institutional agility of the governance model is a relevant factor in minimizing the risks of uneven development that the RIS3 approach can bring to the

maturation of regional innovation systems. In this context, it is recommended that, in the guidelines of the Nacional Coordination Council (Nacional RIS3) and of the governance model of the current framework (Portugal 2020) itself, conditions can be created for governance at the regional level, more strongly articulated with the characteristics of the RIS that frame them, with greater sensitivity to outcome indicators that can better reflect the starting structural conditions in which RIS3 is implemented [78]. In Spain, the governance of RIS3 also had some constraints. It was difficult to integrate participatory governance's novelty in elaborating strategies and their implementation, monitoring and evaluation, with the traditional administration bound to a very strict administrative procedure [76]. In sum, both countries, with their regional and national structures, present difficulties in governance of their smart specialization strategies.

Although the EDP supposedly requires the involvement of a wide range of stakeholders, i.e., representatives from all areas of society (the fourfold helix), instead, in many regions, it is represented by private, public or non-profit sector elites [79]. Additional obstacles include companies' low absorption capacity, limited worker qualifications [67], and/or insufficient university commitment to these policies [80]. This process, which aims to foster internationalizing innovation, thus becomes debilitated or even non-existent. Insufficient U-I links can undermine the innovation system's effectiveness, and consequently the still very nascent EDP. This is perhaps one of the biggest obstacles to the process [73,81–87]. Thus, these results contradicted the fourth proposition (P4: The EDP allows discovering new domains of opportunities).

Despite the assumption that regional innovation is improving, RIS3 did not have the expected impact on Portugal and Spain's performance. Portugal, despite having become a Strong Innovator, has much that remains to be done in terms of smart specialization, as can be seen from the low number of thematic priorities where there has been participation by companies in quest of EU funds to leverage the innovation needed in an increasingly competitive business world.

R&D intensity has decreased since the economic crisis, and despite some changes in Portugal, smart specialization measures have not changed that trend. As pointed out by both the focus group and documentation, if the EDP was the focal point of RIS3, it has failed. This is due both to how the smart specialization strategy was implemented, i.e., the lack of adequate governance models, and the countries under the study's particularities. These are two countries with a history of poor U-I linkage.

### 5.2. Discussion: RIS3 and Open Innovation Dynamics

European research policy underlines the effectiveness of the proximity approach for local university research centres to innovation to local businesses and small firms to improve knowledge and technology transfer [88–90].

This leads us to the topic of open innovation in RIS3 dynamics.

Open innovation is "the inputs and outputs of knowledge to accelerate internal innovation and expand markets for the external use of innovation" [91]. In fact, open innovation is an innovation model based on knowledge inputs and outputs to leverage internal innovation processes, reaching new paths to the market as companies seek to advance their technologies [92]. It is based on this choice of priorities reflected in the RIS3 plan, considering the most relevant areas in each region, allowing the identification of elements that assist in the formulation of strategies, guiding and grounding in business practices. This open innovation concept is a founding element of the types of collaborative relationships monitored in RIS3 and is also very relevant to identifying innovative actors and resources in public and private sectors [93].

According to our results and [76], the Portuguese and Spanish business fabric is mainly made up of micro and SMEs, and community funds, allocated under RIS3, could boost open innovation. Since small companies struggle with technological change as they cannot invest and create the industrial conditions for R&D and deliver the value-added products that the market needs [94–96], the Regional Innovation Strategies could be the mechanism

to boost open innovation, and in this matter corporate innovation. Nevertheless, despite the improvements in terms of innovation in Portugal, there is still, in both countries, a low rate of knowledge and technology transfer from R&D centres to companies, as well as a weak capacity of the industry to incorporate innovation form those centres.

It is important to bear in mind that only enterprises whose activity falls within the predefined priority areas are financed. This intensifies the difficulties experienced by these companies, without intramural innovation, in their search for partnerships that favour their involvement in both national and international markets.

With the implementation of RIS3, the EU intended, however, to fill the existing market failures in this continent, leveraging an entrepreneurial system that allows a business dynamic of open innovation, thus creating networks between the various stakeholders and enabling regional development.

## 6. Conclusions

### 6.1. Theoretical and Empirical Implications

These results allow us to conclude that, despite the need for investment priorities that leverage innovation and, consequently, the European regions' economy, the intelligent specialization strategy has not achieved the expected results. As U-I collaboration depends heavily on community funds, RIS3 can negatively affect already weak links with its impositions and restrictions.

This study contributes in two ways to the literature. Firstly, data are presented on the new rules for the allocation RIS3 community innovation support funds. Secondly, it is shown that the RIS3 did not foster innovation in Portugal and Spain and that the smart specialization strategies require review.

RIS3 aims to leverage the innovation of European countries so that they become internationally competitive. However, we cannot forget the heterogeneity of these countries. Thus, smart specialization strategies should be rethought. Community innovation funding should not be restricted to strongly economic and/or scientific regions, leaving out others where the economic benefit might be greater. For example, priority areas based on endogenous products are often denied funding, not allowing innovation to be funded based on products from other regions or countries. This innovation bottleneck should be reviewed, and all innovation ought to be funded. It should be borne in mind that smart specialization strategies cannot be based on excessively technological innovation, especially in regions with a business fabric consisting mostly of micro and SME. In fact, and according to the Oslo Manual, the exclusion of small innovations contradicts the potential for significant growth from incremental improvements [92].

In terms of open innovation, it is imperative to understand how RIS3 procedures help support this model and if not, where the bottleneck is and which measures could be applied by European Community or the countries.

### 6.2. Policy Recommendations

Portuguese and Spanish innovation still has a long way to improve their innovation performance to become strong innovators in the case of Spain, or leaders in Portugal's case. RIS3 monitoring and governance measures are crucial to providing regions with what they need: efficient use of EU funds that might support business innovation. Regions should look at the results of EU funding and rethink the priorities. It is counterproductive to continue to reject projects if they are proven to be innovative and the products capable of internationalization, simply because they are not aligned with the chosen priority areas. Support measures should be put in place for companies that do not fit into the priority areas now chosen or to be chosen. Such measures could be national, non-EU incentives with a lower funding rate.

This path also requires more effective and efficient U-I collaboration, which might boost EDP implementation. Measures to encourage said collaboration are necessary. On the one hand, we have an entrepreneurial fabric, mostly micro or SMEs, with insufficient

financial or human resources to develop innovation, and much to innovate technologically. On the other hand, we still have very inactive universities looking for industry partnerships, promoting knowledge and technology transfers. Such measures could include a reduction in the tax burden for companies and a premium for universities. U-I linkages should also be reviewed, and (more) specialized structures for seeking partnerships and funding should be set up.

*6.3. Limitations and Future Research*

This study has several limitations: firstly, subjectivity, which is always an issue in qualitative studies, particularly in data analysis via the interview coding and categorization process. Moreover, the focus group lacked any business representatives. Finally, the similarities between the two countries studied should be considered. Future research should compare Portugal and Spain with other countries, both moderately innovative ones and leaders, to determine whether the issues raised here are similar and why. Finally, it would also be useful to look into the impact of RIS3 on this community framework following its completion.

**Author Contributions:** C.M.: Conceptualization, Methodology, Writing—original draft; C.S.M.: validation, funding acquisition, Writing—original draft; J.J.F.: writing—review and editing; A.R.G.: Writing—review & editing. All authors have read and agreed to the published version of the manuscript.

**Funding:** This work is supported by national funds, through the FCT—Portuguese Foundation for Science and Technology under the projects UIDB/04011/2020 and UID/GES/04630/2020.

**Institutional Review Board Statement:** The study did not require ethical approval.

**Informed Consent Statement:** Not applicable.

**Data Availability Statement:** The data presented in this study are available on request from the corresponding author. The data are not publicly available due anonymity of respondents in the foccus group.

**Conflicts of Interest:** The authors declare no conflict of interest.

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
