# Peer review of "The Influence of Research and Innovation Strategies for Smart Specialization (RIS3) on University-Industry Collaboration"

_2199-8531, doi:10.3390/joitmc7010082_

Round 1

Reviewer 1 Report

At first I would like to thank the authors the opportunity of reading their work and providing comments I expect as being helpful towards the improvement of the article. 

The article casts light on an important topic, it is interesting, despite needing some further reflection: 

  1. The introduction starts mentioning the 2008 crisis, I believe that this milestone is dangerous as Portugal was hit with the financial crisis' effect in 2012. And, RIS3 had an horizon of implementation 2014-2020. 
  2. Reference [2] seems obsolete, and findings such as "Moreover, there is no excellence in innovation" seem to be generalistic as the EU encompasses a bunch of the world's innovation leaders. 
  3. Proposition 3 Process governance models are properly implemented and facilitate the needed changes to RIS3. - It is difficult to understand what do the authors refer to here.
  4. Some abbreviations are not explained (e.g EDP)
  5. line 153 I believe is "pick winners"
  6. Please revise the English on your document: In order to triangulate the data, data was collected from each region’s RIS3 documents, statistics from the Regional Innovation Scoreboard, PORDATA, and EC reports, as well as information available on official pages.  - Please details the indicators and the sources this reference does not allow the reader to replicate your figures. 
  7. As the authors know for certain, Portugal is considered, no longer, as a moderate innovator (https://ec.europa.eu/growth/sites/growth/files/eis2020_leader_map-01.png) please revise accordingly. 
  8. It would be very interesting to debate the role of the RIS3 in this transition.
  9. As the option was mentioning the Iberian case (e.g. line 538) it would be interesting to discuss why did Spain remain as a moderate innovator.
  10. In this vein, another angle of the Portuguese reality was analysed in Joana Costa & Carlos Rodrigues, 2020. "Why innovative firms do not rely on universities as innovation sources?," Global Business and Economics Review, Inderscience Enterprises Ltd, vol. 22(4), pages 351-374. Please consider the comparison with their results. 
  11. To me, there is no clear connection about the topic in discussion and Figure 3 and 4, as UIC cannot be addressed in a landscape in which we have the sectors in separate. What the literature tells us is that the researchers need to be bridging knowledge from one dimension to the other.
  12. Please reconsider your affirmation in lines 558, 559 and 560 as accordingly to the EIS that does not seem to adjust to reality (https://ec.europa.eu/commission/presscorner/detail/en/QANDA_20_1150). Please read the report and appraise the strengths and the weaknesses highlighted. The same applies for lines 523-530.
  13. The article should encompass a section clearly explaining the theoretical and practical contributions as well as your policy recommendations based on what the authors think as important to overcome the status quo.
  14. In the final discussion, I believe that the role of both open innovation strategies 

Author Response

Dear editor

Thank you for the opportunity to submit a revised version of our paper to the Journal of Open Innovation: Technology, Market, and Complexity.

In the Word document attached, we explain the changes that have been made to the manuscript to address the comments and suggestions provided. In order to explain explicitly how we have addressed each of the suggestions, we include the text containing each of the Referees’ comments (in italics), followed by an explanation of the revisions undertaken (in regular typeset).

We did our best to implement the revisions suggested by the reviewers. We hope that the revised version of the paper meets your expectations.

Yours sincerely,

The authors

Reviewer 2 Report

Dear Authors,

Thank You for the opportunity of reading this article. My general opinion is positive.

General statements:

-> The article contributes to a better understanding of research and innovation strategies for smart specialization and to empirically assess how they influence university-industry collaboration. So the article's aim is correct and actual. The research in this area are highly recommended.

-> The article content suite to J. Open Innov. Technol. Mark. Complex scope.

-> Abstract is adequate to article content

-> Keywords are correctly proposed

-> Literature background is based on 85 positions, the sufficient number of them are actual

-> The structure of the article is correct.

-> Discussion of the results is deep.

-> Conclusions are correct and clearly indicate the limitations of the proposed solution as well as future research directions.

A minor revision is needed for more technical than methodology issues. They are as follows:
Figures quality is poor. Please revise figures 2, 3, 4.

Please add DOI to references, where they are available.

Author Response

Dear Professor JinHyo Joseph Yun

Thank you for the opportunity to submit a revised version of our paper to the Journal of Open Innovation: Technology, Market, and Complexity.

In the document attached, we explain the changes that have been made to the manuscript to address the comments and suggestions provided. In order to explain explicitly how we have addressed each of the suggestions, we include the text containing each of the Referees’ comments (in italics), followed by an explanation of the revisions undertaken (in regular typeset).

We did our best to implement the revisions suggested by the reviewers. We hope that the revised version of the paper meets your expectations.

Yours sincerely,

The authors

Reviewer 3 Report

The paper analyses an important topic, which is the relation between university and industry from the perspective of collaboration. It was a pleasure to read it and review it.

I will begin with some minor technical issues:

- Figure 2 uses too small characters that makes it hard to read. In the same time, some acronyms used in the figure should be explained in the text, for instance “HEI”.

- The same applies for the acronym “SMEs” on line 215, even though all these acronyms are well known.

The article contains a lot of literature references and quotes which is a positive thing, but my main concern is that the paper is overall a little too theoretical, while the practical component is minimized and somehow diluted. A more pragmatically approach would be helpful in order to increase the overall value of the research. From this perspective, the Conclusion section seems to be a little too general and should better emphasize the practical implications of the findings. Perhaps inserting the main ideas from Discussion and Conclusion in a brief and synthetic table can help in this regard.

Author Response

(The authors gave the same response as above.)

Round 2

Reviewer 1 Report

Many thanks for taking into consideration the recommendations provided. 

The manuscript has improved significantly. My single concern has to do with some minor typos and some English revisions

All the best.

Author Response

Thank you for your comments. We are pleased to have responded to all your comments and that they were as expected.  

Thank you for your comments. We found the comment relevant and we have made an English revision, namely in lines 9, 144, 152, 177-179, 284, 464-465, 505, 558, 581, 614, 623, 664, 688 and 702.

In the document attached, you can find all the changes we have made. 

Once again, thank you for your comments. 
